# *mRI*: Multi-modal 3D Human Pose Estimation Dataset using mmWave, RGB-D, and Inertial Sensors

**Sizhe An**
University of Wisconsin-Madison
sizhe.an@wisc.edu

**Yin Li**
University of Wisconsin-Madison
yin.li@wisc.edu

**Umit Ogras**
University of Wisconsin-Madison
uogras@wisc.edu

## Abstract

The ability to estimate 3D human body pose and movement, also known as human pose estimation (HPE), enables many applications for home-based health monitoring, such as remote rehabilitation training. Several possible solutions have emerged using sensors ranging from RGB cameras, depth sensors, millimeter-Wave (mmWave) radars, and wearable inertial sensors. Despite previous efforts on datasets and benchmarks for HPE, few dataset exploits multiple modalities and focuses on home-based health monitoring. To bridge this gap, we present *mRI*[1], a multi-modal 3D human pose estimation dataset with **m**mWave, **R**GB-D, and **I**nertial Sensors. Our dataset consists of over 160k synchronized frames from 20 subjects performing rehabilitation exercises and supports the benchmarks of HPE and action detection. We perform extensive experiments using our dataset and delineate the strength of each modality. We hope that the release of *mRI* can catalyze the research in pose estimation, multi-modal learning, and action understanding, and more importantly facilitate the applications of home-based health monitoring.

## 1 Introduction

3D Human pose estimation (HPE) refers to detecting and tracking human body parts or key joints (e.g., wrists, shoulders, and knees) in the 3D space. It is a fundamental and crucial task in human activity understanding and movement analysis with numerous application areas, including rehabilitation [40, 31, 7, 6], professional sports [35], augmented/virtual reality, and autonomous driving [28]. In particular, human pose estimation plays an increasingly important role in healthcare applications, such as remote rehabilitation training [37, 19]. The current mainstream rehabilitation treatment involves a physical therapist supervising the patients in person. In contrast, HPE-based health monitoring systems can help clinicians correct patients' movements or instruct them remotely. To this end, multiple datasets have studied HPE with health-related physical movements [6, 40, 31, 7].

Many existing studies rely heavily on processing RGB frames from color cameras for human pose estimation [20, 5, 16, 34, 17, 25]. RGB image and video frames are the most common input types since they offer an non-invasive approach for HPE. However, the image quality depends heavily on the environmental setting, such as light conditions and visibility [3]. Moreover, using image and video data poses significant privacy concerns, especially in a household environment. Finally, the data-intensive nature of real-time video processing requires computationally powerful equipment with high cost and energy consumption.

---

[1]Project page: http://sizhean.github.io/mri

36th Conference on Neural Information Processing Systems (NeurIPS 2022) Track on Datasets and Benchmarks.

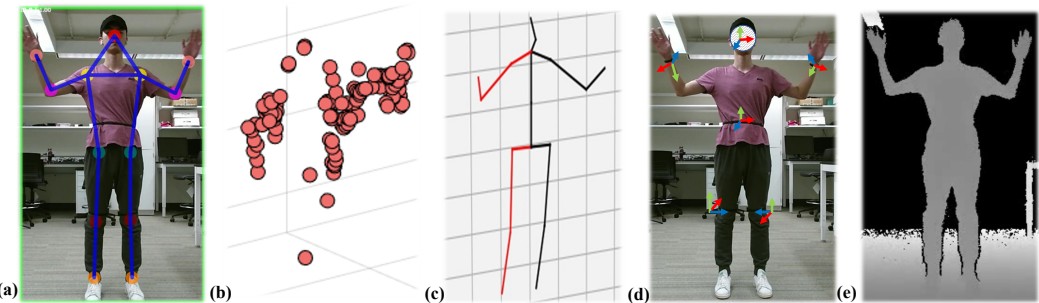

Figure 1: Overview of all modalities and annotations in *mRI* dataset. All sub-figures uses the same sample frame during 'both upper limb extension'. (a) 2D human keypoints with bounding box on RGB image, (b) 3D mmWave point cloud, (c) 3D human skeletons, (d) IMU rotations, (e) depth image. *mRI* dataset supports **human pose estimation** and **action detection** tasks. With *mRI*, researchers from the fields of machine learning, computer vision, wearable computing can exploit the complementary advantages of **multi-modality**, while clinical and rehabilitation experts can focus on its **healthcare** movements.

Frame quality, privacy, and computational power drawbacks of video processing can be addressed by emerging *complementary sensor modalities*, such as lidar, millimeter wave (mmWave) radar [3, 46, 49], and wearable inertial sensors [43, 41, 42, 44, 30, 47, 2], The point cloud from lidar overcomes frame quality and privacy challenges. However, it has a high cost and computation power requirements to process the data, making it unsuitable for indoor applications such as rehabilitation. In contrast, mmWave radar can generate high-resolution 3D point clouds of objects while maintaining low cost, privacy, and computational power advantages. Similarly, wearable inertial sensors provide accurate rotation and acceleration information regarding joints with low cost and computational power requirements [41, 42, 44, 2], yet at a price of body worn sensors.

High-quality and large-scale datasets provide a vital foundation for algorithm development. To catalyze research in HPE, this work (*mRI*) combines mmWave radar, RGB-Depth (RGB-D), and Inertial sensors to exploit their complementary advantages. We present a comprehensive 3D human pose estimation dataset performed by 20 human subjects, consisting of more than 160k synchronized frames from three sensing modalities. The contributions and unique aspects of *mRI* are as follows:

- **Multiple Sensing Modalities.** *mRI* consists of mmWave point cloud, RGB frames, depth frames, and inertial signals. The experimental data is captured using a commercial low-power, and a low-cost mmWave radar, two depth cameras, and six high-accuracy inertial measurement units (IMUs). All sensors are temporally synchronized and spatially calibrated. To the best of our knowledge, *mRI* is the first dataset that combines these complementary modalities, as elaborated in Section 2.

- **Healthcare Movements Focus.** We use ten clinically-suggested rehabilitation movements that involve the upper body, lower body, and the major muscles related to human mobility, as described in Section 3.2. These movements are crucial for patients to recover from sequelae of central nervous system disorders, such as Parkinson's disease (PD) and cerebrovascular diseases (e.g., stroke). Hence, the *mRI* dataset can serve as a reference from healthy subjects, while the experimental methodology can enable future studies with patients.

- **Flexible Data Format and Extensive Benchmarks.** We release the raw synchronized and calibrated sensor data and a comprehensive set of benchmarks for 2D/3D human pose estimation and action detection using multiple modalities (see Section 4). The proposed end-to-end pipeline pre-processes the raw data into the point cloud, features, and 2D/3D keypoints. In addition, all manually-labeled actions annotations and 3D human key points ground truth are released to public, as detailed in Section 3.2.

- **Low-Power & Low-Cost Requirements.** Widespread use of home-based rehabilitation depends critically on the affordability and operating cost of the deployed systems (see Section 3.1). Our *mRI* dataset and findings pave the way to sustainable systems with low-power and low-cost sensors and edge devices. For example, only mmWave radar and IMU sensors can be used in the field after they are trained with all three modalities (including RGB-D) in a clinical environment.

| Dataset | Sensing Modalities | | | | # of Subjects | # of Seqs | # of Actions | # of Synced Frames | Annotations | | |
|---|---|---|---|---|---|---|---|---|---|---|---|
| | RGB | Depth | IMU | mmWave | | | | | Action | 2DKP | 3DKP |
| COCO [20] | ✓ | - | - | - | - | - | - | 104k | - | ✓ | - |
| MPII [5] | ✓ | - | - | - | - | 24k | 410 | 25k | ✓ | ✓ | - |
| MPI-INF-3DHP [25] | ✓ | - | - | - | 8 | 16 | 8 | 1.3M | - | ✓ | ✓ |
| Human3.6M [16] | ✓ | ✓ | - | - | 11 | 839 | 17 | 3.6M | ✓ | ✓ | ✓ |
| CMU Panoptic [17] | ✓ | ✓ | - | - | 8 | 65 | 5 | 154M | - | ✓ | ✓ |
| NTU RGB+D [34] | ✓ | ✓ | - | - | 40 | 56k | 60 | 4M | ✓ | ✓ | ✓ |
| 3DPW [41] | ✓ | - | ✓ | - | 7 | 60 | - | 51k | - | ✓ | ✓ |
| MPI08 [30] | ✓ | - | ✓ | - | 4 | 24 | 24 | 14k | - | - | ✓ |
| TNT15 [43] | ✓ | - | ✓ | - | 1 | - | 5 | 14k | ✓ | - | ✓ |
| MoVi [12] | ✓ | - | ✓ | - | 90 | 1044 | 21 | 712k | ✓ | ✓ | ✓ |
| RF-Pose [50]† | ✓ | - | - | ✓ | 100 | - | 1 | - | ✓ | ✓ | - |
| RF-Pose3D [49]† | ✓ | - | - | ✓ | >5 | - | 5 | - | ✓ | ✓ | - |
| mmPose [33]† | - | - | - | ✓ | 2 | - | 4 | 40k | ✓ | - | ✓ |
| mmMesh [46]† | ✓ | - | - | ✓ | 20 | - | 8 | 3k | ✓ | - | ✓ |
| MARS [3] | - | - | - | ✓ | 4 | 80 | 10 | 40k | ✓ | - | ✓ |
| Reiss et al. [31] | - | - | ✓ | - | 9 | - | 18 | 3.6M | ✓ | - | - |
| HPTE [7] | ✓ | ✓ | - | - | 5 | 240 | 8 | 100k | ✓ | - | ✓ |
| EmoPain [8] | ✓ | - | - | - | 50 | - | 11 | 33k | ✓ | - | ✓ |
| AHA-3D [6] | ✓ | - | - | - | 21 | 79 | 4 | 170k | ✓ | - | ✓ |
| UI-PRMD [40] | ✓ | - | - | - | 10 | 100 | 10 | 60k | ✓ | - | ✓ |
| *mRI* | ✓ | ✓ | ✓ | ✓ | 20 | 300 | 12 | 160k | ✓ | ✓ | ✓ |

Table 1: Comparison across related datasets. For 2D keypoint annotations, only COCO [20] and MPII [5] are annotated manually, all others are derived from deep models. −: Not report in the paper. †: The dataset is not open-source/available. The first group of rows shows earlier RGB and RGB-D datasets. The middle group of rows presents datasets with emerging sensing modalities such as IMUs and mmWave. The last group of rows lists healthcare-related datasets.

## 2 Related Work

### 2.1 3D Human Pose Estimation

Marker-based optical motion capture (MoCap) systems are often used to acquire accurate 3D body pose [16, 8, 40]. Optical MoCap systems require attaching reflective markers to the body and are quite costly, thus are limited to laboratory settings. Recently, MoCap systems based on body-worn IMUs have been developed [41, 30, 44, 43, 12]. They are considerably cheaper yet at a cost of tracking accuracy due to drifting [1]. Our dataset explores using low-cost IMUs for 3D HPE.

Besides marker-based MoCap, Marker-less MoCap has received much attention. Depth cameras are often used for pose estimation [26], yet are limited by their sensing range (within 5 meters). Recent effort has focused on pose estimation using RGB cameras. With the help of machine learning, 3D joints can be estimated from a single RGB image [23], or from several RGB images from different viewing angles captured by multiple cameras [17, 25, 30], or from a sequence of RGB frames within a video [29]. However, RGB cameras are easily affected by poor light conditions, and raise privacy concerns for home-based monitoring. More recently, mmWave-based pose estimation, including radio frequency sensing, has emerged as a promising solution [50, 49, 33, 46, 3]. A mmWave-based solution has demonstrated comparable accuracy to RGB and depth cameras, yet excels at privacy-preserving and long working range. Our dataset includes mmWave for 3D HPE.

Moving forward, the results of 3D HPE can be used by skeleton-based action recognition [22, 34] to localize and recognize actions in time, broadening its applications in health monitoring [21, 27] and human behavior analysis [32]. Our dataset provides action annotations and we evaluate using the estimated pose for temporal action localization [11].

### 2.2 Datasets for Human Pose Estimation

High-quality datasets with annotations are crucial for the advancement of pose estimation. Table 1 summarizes previous works on HPE datasets and compare them to our *mRI*. Some of the early

effort focuses on 2D HPE (e.g., COCO [20] and MPII [5]), or 3D HPE a single modality (e.g., 3DHP with images, mmPose [33] with mmWave, and MPI08 [30] with IMU). More recent works combines multiple modalities for 3D HPE. For example, Human3.6M [16] contains RGB images and depth maps of 11 professional actors performing 17 daily activities, coupled with ground-truth 3D poses from optical MoCap. RF-Pose3D [49] presents the first study to use radio frequency sensing for 3D HPE, together with a dataset of both RGB images and radio signals. MoVi [12] incorporates both IMU signals and RGB frames, as well as ground-truth 3D poses from MoCap, and presents a benchmark for both 3D HPE and human activity recognition. In comparison to existing dataset,

To the best of our knowledge, *mRI* is the *first HPE dataset with the most comprehensive set of sensing modalities*, including RGB, depth, IMU, and mmWave. In addition, *mRI fills the vacancy of standardized mmWave-based human pose estimation*, as all current mmWave-based HPE datasets are either not open-sourced or without proper keypoints annotations and RGB references.

### 2.3 Human Pose Estimation for Rehabilitation

HPE promises to capture complex body movement naturally occurring in daily activities or prescribed by clinicians, and thus offers a promising vehicle to inform treatment and to quantify the progress of treatment. Individual sensing modality has been previously considered, including RGB camera [7, 8], depth camera [6, 18, 40], IMUs [31], and MoCap [8, 40]. Reiss et al. [31] presents a dataset monitoring physical activities with three IMUs and a heart rate monitor. The home-based physical therapy exercises (HPTE) dataset [7] uses Kinect to record video and depth streams while users perform eight therapy actions. The EmoPain dataset [8] captures both joint information and face videos to classify the pain level based on the emotion in the rehabilitation movements. The AHA-3D [6] dataset contains 79 skeleton videos recorded by Kinect for four healthcare activities. Similarly, the UI-PRMD [40] dataset captures common physical rehabilitation exercises using the Kinect and Vicon MoCap. Similar to these works, *mRI* focuses on rehabilitation exercises, and provides the most comprehensive set of sensing modalities while remaining competitive in its scale.

## 3 Dataset

*mRI* includes 3D point cloud from mmWave, RGB frames and depth maps from RGB-D cameras, joints rotations and accelerations from wearable IMU sensors, as well as annotations of 2D keypoints, 3D joints, and action labels of 12 clinically relevant movements. *mRI* consists of 300 time-series sequences with 160K synchronized frames and more than 5M total data points from all sensors, from 20 subjects. We hope that our dataset will contribute to the multi-modal machine learning community, and facilitate applications of HPE for rehabilitation and other healthcare problems. In what follows we describe the hardware system to capture the data and the data collection process. More details can be found in the supplementary A.3.

### 3.1 Capturing Multi-Modal Signals for Human Pose Estimation

To capture multi-modal data, we designed a sensor system composed of one mmWave radar, two sets of RGB and depth cameras, and six wearable IMUs. Detailed specifications and features of all sensors are shown in Table 2. The mmWave radar and two Kinect V2 sensors are placed on a desk 2.4 m away from the subject, wearing six IMU sensors, as shown in Figure 2. We now describe the data capturing for each modality and the synchronization across modalities.

| | # | Freq. | Con. | Power | Privacy↑ | Anti-inter.↑ | Intrusive | Output |
|---|---|---|---|---|---|---|---|---|
| mmWave [38] | 1 | 10 Hz | Wired | 2.1 W | ★★★ | ★★★ | No | Point cloud |
| RGB [26] | 2 | 30 Hz | Wired | 16 W | ★ | ★ | No | RGB frame |
| Depth [26] | 2 | 30 Hz | Wired | 16 W | ★★ | ★★ | No | Depth and infra-red frame |
| IMU [45] | 6 | 50 Hz | BLE | 120 mW | ★★★ | ★★★ | Yes | Accelerations and quaternions |

Table 2: **Comparison across sensors**. #: Number of sensors. Freq.: Sampling frequency. Con.: Type of connection to the host PC. Privacy indicates privacy-preserving ability. Anti-interference represents how much it is affected by environmental factors like non-ideal lighting.

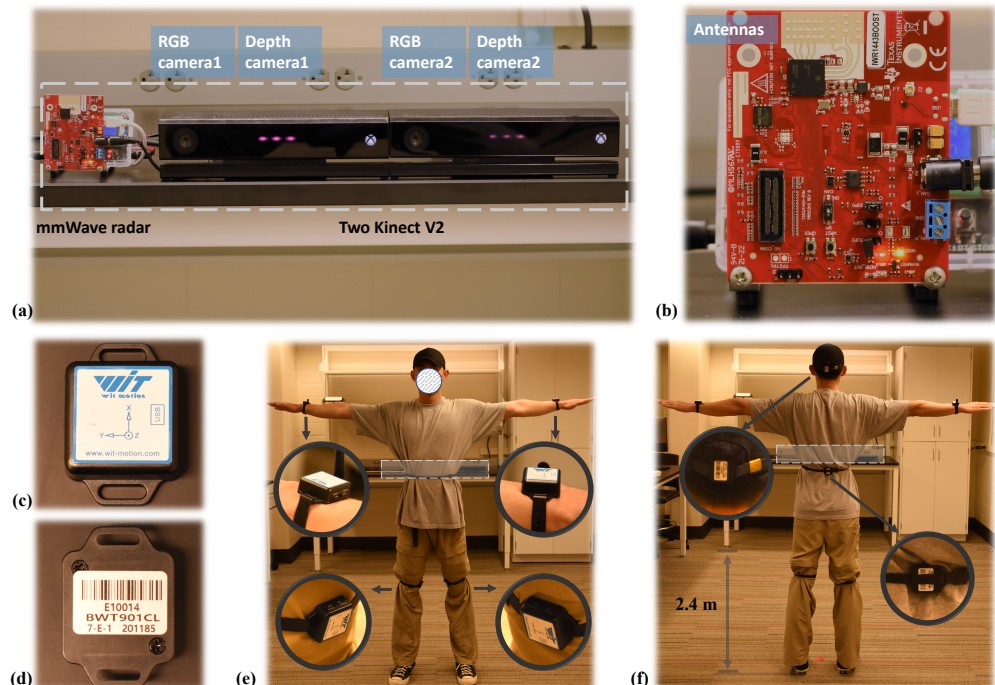

Figure 2: Overview of the experimental setup. (a) shows all non-intrusive sensors, including mmWave radar, two RGB, and depth cameras. (b) shows a zoom-in version of the mmWave radar and its antennas. The front and back views of the IMU are shown in (c) and (d), respectively. (e) and (f) show the front and back view of the subject standing as a "T pose" with six IMUs and zoom-in views of IMUs. The gray dash line boxes in (a), (e), and (f) represent the position of non-intrusive sensors.

**Point cloud from mmWave radar**. A Texas Instruments (TI) IWR1443 Boost mmWave radar [38] is used to obtain the mmWave point cloud. 3D mmWave point cloud is generated by Frequency Modulated Continuous Wave (FMCW) radar using multiple transmit (Tx) and receiver antennas (Rx) configuration [3, 33, 49]. The radar emits a chirp signal, a sinusoid wave whose frequency increases linearly with time. Then the reflected signals are received at the Rx antenna side. The range, velocity and angle resolutions are computed with the received data using *range FFT*, *Doppler FFT*, and *angle estimation* algorithms, respectively. After the constant false alarm rate (CFAR) algorithm eliminates the noise, a point cloud capturing object shape and movement is constructed as

$$P_i = \big(x_i, y_i, z_i, d_i, I_i\big), i \in \mathbb{Z}, 1 \le i \le N \tag{1}$$

where $x_i, y_i, z_i$ are the spatial coordinates of the point, $d_i$ represents the Doppler velocity, $I_i$ denotes the signal intensity, and $N = 64$ represents the total number of points in a given frame. To further increase the density of the point cloud, we follow [4] to fuse points from three consecutive frames, i.e., increasing the number of points per frame from 64 to 192. See more detalis in the Supplement A.4.

The radar is connected to the host PC through the UART interface. We modify a Matlab Runtime implementation from TI [39] for the data acquisition. The sampling rate is set to 10 Hz since it is sufficient for measuring human movement (the frequency of most voluntary human movements spans from 0.6 to 8 Hz [13]).

**RGB and depth frames from RGB-D cameras**. Two Microsoft Kinect V2 [26] sensors are used to capture RGB and depth frames. Kinect V2 has a high precision color camera and infra-red camera, generating color and depth frame with a resolution of $1920 \times 1080$ and $512 \times 424$, respectively. We modified the software from libfreenect2[2] to generate aligned color, depth, and infra-red frames with the global timestamp from two Kinect V2 sensors. We calibrate the two cameras using the Matlab camera calibration toolbox [24]. The center of the RGB camera 1, as shown in Figure 2 (a) is selected as the origin of the world coordinate system.

[2] https://github.com/OpenKinect/libfreenect2

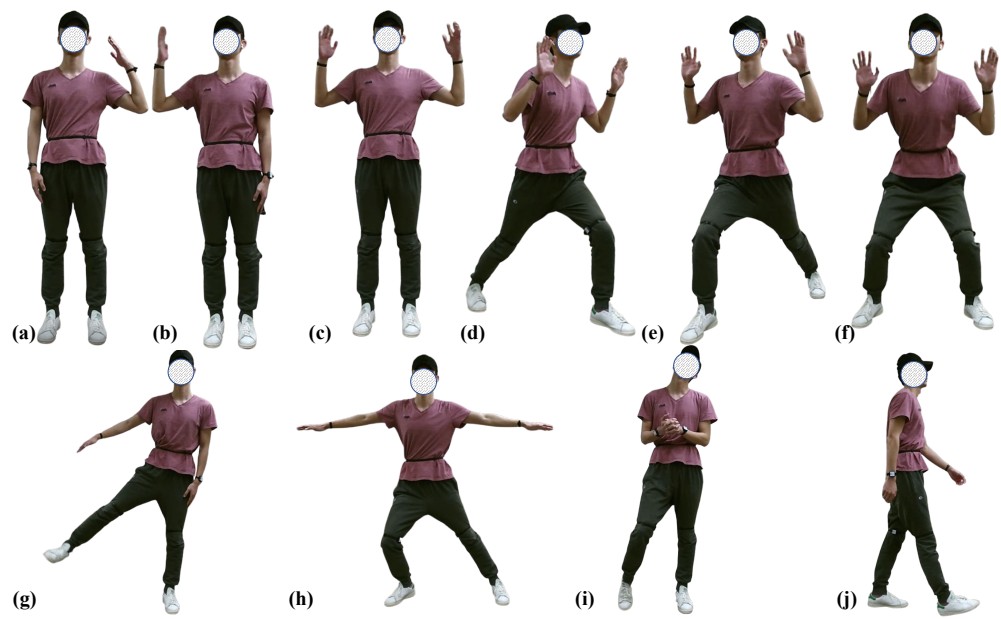

Figure 3: Overview of all movements in *mRI*, as described in Section 3.2. The mirror movements of (g) and (h) are not shown due to limited space.

**Joints rotations and accelerations from wearable IMUs**. Six Wit-motion BWT901CL IMUs [45] are used to capture the rotation and acceleration of the human joints. In our experiments, the IMUs are tightly attached to the left wrist, right wrist, left knee, right knee, head, and pelvis of the subject to capture the complete information about the human body, as shown in Figure 2(e) and (f). Each IMU contains a 3-axis accelerometer, 3-axis gyrometer, and 3-axis magnetometer as the sensing unit. The raw output data from the sensors are accelerations, angular velocity, Euler angle, and magnet field values. Based on these values, we extract joint quaternion and 3-axis acceleration following [15, 47] as they fully specify the body pose and movement. The IMUs connect to the host PC via a USB-HID device using the BLE protocol with a sampling rate of 50 Hz (see Table 2), ensuring low-latency data transmission with multiple devices.

**Sensors synchronization**. All sensors are connected to the same host PC, allowing global timestamps from the host attached to each data point from different sensors. We then synchronize all data points using these global timestamps. Since mmWave radar has the lowest sampling rate, we use its timestamp as the basic timestamp. For each timestamp in mmWave radar, we find the timestamp in other sensors with the minimum absolute difference between itself and the mmWave timestamp and align them. The time difference between sensors is less than 5 ms with the proposed time alignment method. Finally, the synchronized data across all modalities have the same number of data points.

### 3.2 Data Collection, Annotation, and Visualization

**Rehabilitation exercises**. We consider 12 movements related to rehabilitation exercises covering the entire human body. The first ten rehabilitation movements are modified from [40, 3]. Figure 3 shows all movements: (a) left upper limb extension, (b) right upper limb extension, (c) both upper limb extension, (d) left front lunge, (e) right front lunge, (f) squat, (g) left side lunge, right side lunge, (h) left limb extension, and right limb extension. The $11^{th}$ and $12^{th}$ movement are stretching and relaxing in free forms (i), and walking in a straight line (j), respectively. These two movements are meant to increase the diversity of the dataset, as the $11^{th}$ movement is determined by each subject and the $12^{th}$ movement features a global displacement. The duration of each type of movement is around one minute per subject. To calibrate the IMUs, we require the subject to perform a "T Pose" at the beginning of each recording.

**Participant recruitment and consent**. To conduct human subject study, we obtained an approval from the IRB at the university. Our participants were recruited locally and all experiments were

carried out in a laboratory setting. Before each session, a researcher introduces the research goal, experiment procedure, and potential risk via both verb communication and video tutorials. The participant is free to raise questions before he or she sign the consent form, and is free to withdraw from the study at any time. We refer more details to our Ethic statements.

20 healthy participants consented and managed to perform the study. There are 13 males and 7 females, with an average age of 24.1±4.4 and a height of 175.6±9.3 cm.

**Obtaining human body pose**. We now describe how we derive 2D keypoints and 3D joints given our sensor data. Without using MoCap, our solution is a combination of 2D keypoint detection (body parts), 3D triangulation (joints), and an optimization-based refinement.

- First, we use HRNet [36] (with bounding boxes from Mask RCNN [14]) to detect 2D keypoints of human body parts in all RGB frames from both cameras.
- Next, we triangulate two sets of 2D keypoints captured at the same time yet from different cameras, using camera parameters obtained via camera calibration. The results are a set of 3D body joints (17 in total following COCO format).
- Finally, we refine the 3D joints in each video by solving an optimization problem. Our optimization minimizes 2D reprojection error, imposes equal bone length constraint for all frames, and enforces temporal smoothness of the 3D joints.

Specifically, our refinement step solves the following optimization problem

$$\min_{\{\mathbf{p}_i\}} \sum_{i=1}^{\mathbb{Z}} \left( \left\| P^l \mathbf{p}_i - \mathbf{q}_i^l \right\| + \left\| P^r \mathbf{p}_i - \mathbf{q}_i^r \right\| \right) + \sum_{j}^{bonelist} \left\| \mathbf{B}_j - median(\mathbf{B}) \right\| + \sum_{i=1}^{\mathbb{Z}-1} \left\| \mathbf{p}_{i+1} - \mathbf{p}_i \right\|, \quad (2)$$

where $\{\mathbf{p}_i\}$ is the set of 3D joints of size $\mathbb{Z}$, $\mathbf{q}_i^l$ and $\mathbf{q}_i^r$ are the 2D keypoints from the left and right camera, respectively. $P^l$ and $P^r$ are the camera projection matrix for the left and right camera, respectively. $\{\mathbf{B}_j\}$ is a set of bone length defined by connecting a subset of the joints (e.g., wrist to elbow, elbow to shoulder). The first term represents the re-projection errors of the two cameras. The second term enforces equal bone length across all frames in the same video (i.e., the same subject). And the third term imposes temporal smoothness of the 3D joint coordinates. More details, including both quantitative and qualitative results, can be found in the supplement. After the optimization, we re-project the 3D joints to 2D and thus update the 2D keypoints.

**Keypoints quality**. To validate the reliability of the obtained 3D joints, we report the reprojection error of the derived 3D joints by comparing their 2D projections to human annotated 2D keypoints. Specifically, we randomly sample 50 video frames from our dataset, manually annotate the 2D keypoints for each frame, and calculate the error between the projected 3D joints and the annotated 2D keypoints, following [5]. The mean absolute percentage error (MAPE) is 1.5%, and the percentage of correct keypoints thresholded at 50% of the head segment length (PCKh) is 98.9. More details and visualization can be found in the supplement A.2.

**Annotating actions in videos**. We also provide annotations of the 12 movements for each video. The multi-media annotation tool ELAN [9] is employed to annotate the videos. For each video sequence, we manually label the start and end timestamp and the category of the 12 different movements.

## 4 Evaluation and Benchmarks

We introduce a standardized evaluation pipeline of using our dataset for 3D human pose estimation and human action detection. We use latest models to benchmark the performance of each modality and discuss their results.

### 4.1 3D Human Pose Estimation

Our main benchmark is 3D HPE. We now describe our experiment protocol, evaluation metrics, and the method we used, followed by the presentation of our results.

**Experiment protocol**. We consider two settings of data splits. **Setting 1 (S1 Random Split)**: A random split of 80% and 20% of all data is used as the training and testing set, respectively. **Setting 2 (S2 Split by Subjects)**: A randomly selected subset (80%, i.e., 16 out of 20) of the subjects is

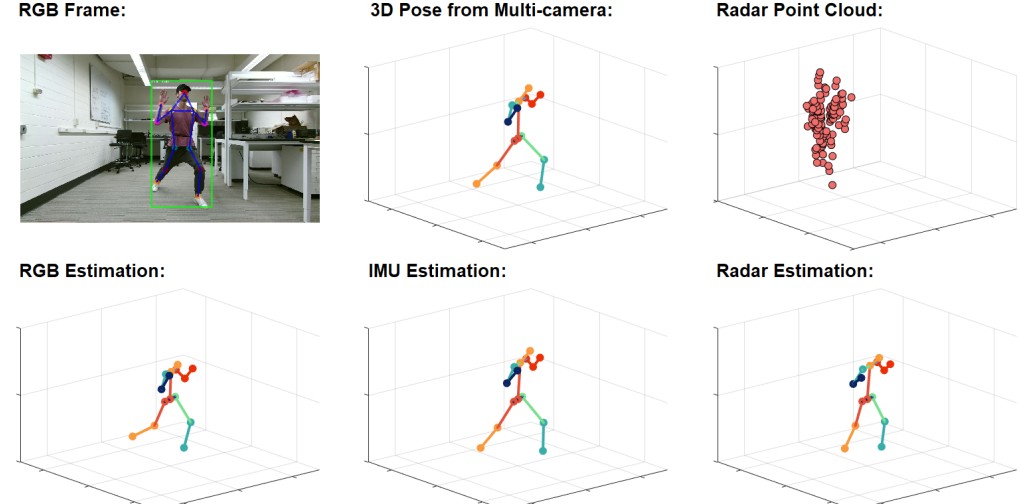

Figure 4: Visualization of sample pose data and results during left front lunge. Top row (from left to right): an RGB frame with detected human bounding box and 2D keypoints, the refined 3D pose derived from two cameras, and the 3D point cloud from mmWave radar. Bottom row (from left to right) shows the estimated 3D pose from a single RGB camera, IMU signals, and mmWave radar.

used for training, while the rest are for testing. S1 mimics a case where personalized HPE model is possible, while S2 evaluates across-subject generalization. For each setting, we randomly sample three splits and report the averaged results. More details are provided in the supplement A.3.

Further, we also define two evaluation protocols based on the design of our movements, as mentioned in Section 3.2. **Protocol 1 (P1)** consists of all 12 movements, including stretching and relaxing in free forms and walking. While **Protocol 2 (P2)** only considers the first ten rehabilitation movements. Such protocols help us investigate the robustness of the model in terms of fixed/free form movement.

**Evaluation metrics**. We adopt Mean Per Joint Position Error (MPJPE) and Procrustes Analysis MPJPE (PA-MPJPE), widely used in human body pose estimation [16], as the main metrics. MPJPE represents the mean Euclidean distance between ground truth and prediction for all joints. MPJPE is calculated after aligning the root joints (the pelvis) of the estimated and ground truth 3D pose. PA-MPJPE is MPJPE after being aligned to the ground truth by the Procrustes method [10], a similarity transformation including rotation, translation, and scaling. We also report additional metrics such as joint angles provided in the supplement.

**Methods**. We conduct 3D human pose estimation using mmWave, RGB, and IMUs separately using latest methods. Here we briefly introduce the methods considered in our evaluation and refer to our supplement for more implementation details.

- **mmWave**: We use the data processing pipeline and model from [3] that learns a convolutional neural network on the 5D point cloud to regress the 3D joints. The model is trained from scratch on our dataset, and outputs the 3D joints in the global coordinates system.

- **RGB**: We adopt the model from [29], where 2D keypoints from a sequence of frames are "lifted" into 3D joints (in the camera coordinate system) using a convolutional neural network. We use the pre-trained model from [29]. As the pre-trained model outputs a different set of joint, we only evaluate on a subset that intersects with our set of joints.

- **IMUs**: We employ the feature processing method from [47], with a convolutional neural network trained to regress rotations relative to a root joint (e.g., pelvis) using data from IMUs. The model is trained from scratch on our dataset.

**Results and discussion**. Table 3 shows the 3D HPE results for mmWave, RGB, and IMUs. Under **S1** and **P1**, mmWave-based HPE achieves 163 and 94 mm for MPJPE and PA-MPJPE, respectively. The metrics are further reduced to 125 and 74 mm for **P2**. IMU-based HPE obtains MPJPE and

| Modality | Setting | Protocol 1 | | Protocol 2 | |
|---|---|---|---|---|---|
| | | MPJPE (mm)↓ | PA-MPJPE (mm)↓ | MPJPE (mm)↓ | PA-MPJPE (mm)↓ |
| mmWave | S1 | 163.3±9.1 | 94.1±3.6 | 125.1±2.4 | 74.1±1.0 |
| | S2 | 186.6±23.8 | 97.3±7.8 | 126.6±11.3 | 75.0±7.1 |
| RGB | S1 | 116.9±0.1 | 66.8±0.2 | 115.0±0.1 | 64.4±0.1 |
| | S2 | 120.1±3.7 | 67.5±1.9 | 118.4±3.8 | 64.7±1.4 |
| IMUs | S1 | **80.2**±12.6 | **51.9**±1.9 | **40.9**±1.0 | **28.4**±0.9 |
| | S2 | 147.4±18.4 | 74.5±5.9 | 94.3±13.8 | 54.0±4.9 |

Table 3: 3D human pose estimation results for mmWave, RGB, and IMUs. We report the mean and standard deviation of joint errors averaged across multiple splits under both our settings (**S1** & **S2**).

PA-MPJPE of 87 and 60 mm, respectively, under **S1** and **P1**. Figure 4 shows visualization comparison of estimation across different modalities.

Under **S2**, mmWave-based HPE performs similarly to **S1**, while IMU-based HPE obtains worse results than **S1**. This is because the sensing data from IMU is more fine-grained on the joints while mmWave grasps more information about body trunk, which is not too subject-specific. As a result, the IMU-based model is more sensitive to different subjects. We can observe that for all modalities **S2** yields higher standard deviations than **S1** since the difference between subjects is much more significant than random split, between train and test set. Similarly, **P1** yields higher standard deviations than **P2** since all movements in **P2** are fixed positions, which makes the model learning the keypoints distribution easier.

RGB-based HPE achieve 116 and 66 mm MPJPE and PA-MPJPE for **P1** under **S1**. Both data-split yield similar results. To compare, the same model achieves 36 mm PA-MPJPE on Human3.6M dataset. However, the model is trained and evaluated on Human3.6M while it is only evaluated on *mRI* without any fine-tuning. We leave fine-tuning the model on *mRI* as future work. In summary, all modalities perform reasonably well on our dataset.

**Result visualization**. We further visualize sample results of 3D pose estimation from different modalities in Figure 4. Additional examples can be found in the supplement A.5.

## 4.2 Skeleton-based Action Detection

Moving forward, we explore using the estimated 3D joints for temporal action detection in untrimmed videos — the simultaneous localization and recognition of action instance in time. Specifically, given an input untrimmed video, temporal action localization seeks to predict a set of action instances with varying size. Each instance is defined by its onset, offset, and action labels.

**Experiment protocol**. We consider the more challenging setting **S2**, where a model is tasked to detect actions performed by subjects not presented in the training set. Here each movement type defines one action category. Similar to our HPE experiments, we evaluate on both **P1** (12 categories) and **P2** (10 categories focusing on rehabilitation exercises). Importantly, we consider using individual modalities and all combinations of these modalities (e.g., RGB+IMU or RGB+mmWave). To combine multiple modalities, 3D joint data from each modality at every time step is concatenated, and the resulting sequence is fed into the model.

**Evaluation metrics**. Following prior work [11], we report the mean average precision (*m*AP) at multiple temporal intersection over union (tIoU) thresholds ([0.5:0.05:0.95]). tIoU is defined as the intersection over union between two temporal windows, i.e., the 1D Jaccard index. Given a tIoU threshold (e.g., 0.75), *m*AP computes the mean of average prevision across all action categories. An average *m*AP is also reported by averaging the *m*AP scores across all tIoUs.

**Method**. We make use of a latest method — ActionFormer [48] for temporal action detection. ActionFormer develops a Transformer based model and achieves state-of-the-art results across action detection benchmarks. Specifically, we feed the model with a sequence of estimated 3D poses from different modalities at a sampling rate of 2 Hz, and train the model from scratch on our dataset.

**Results and discussion**. Table 4 summarizes the results from three modalities and their combinations averaged across all splits. Overall, all modalities perform fairly well, with *m*AP scores around 90%. Under **P1**, IMUs data have the best results with 93.4% *m*AP, and outperform the RGB frames and

| Modality | Protocol 1 | | | | Protocol 2 | | | |
|---|---|---|---|---|---|---|---|---|
| | tIoU=0.50 | tIoU=0.75 | tIoU=0.95 | average | tIoU=0.50 | tIoU=0.75 | tIoU=0.95 | average |
| mmWave (**W**) | $98.22_{\pm3.08}$ | $97.59_{\pm4.17}$ | $29.02_{\pm6.31}$ | $87.04_{\pm4.89}$ | $99.00_{\pm1.73}$ | $97.21_{\pm2.41}$ | $31.11_{\pm15.34}$ | $87.55_{\pm3.61}$ |
| RGB (**R**) | $\mathbf{100.00}_{\pm0.00}$ | $99.14_{\pm0.75}$ | $44.80_{\pm10.55}$ | $91.56_{\pm2.08}$ | $\mathbf{100.00}_{\pm0.00}$ | $\mathbf{100.00}_{\pm0.00}$ | $59.87_{\pm8.12}$ | $\mathbf{95.07}_{\pm1.46}$ |
| IMUs (**I**) | $\mathbf{100.00}_{\pm0.00}$ | $\mathbf{100.00}_{\pm0.00}$ | $53.55_{\pm12.39}$ | $\mathbf{93.46}_{\pm2.30}$ | $\mathbf{100.00}_{\pm0.00}$ | $\mathbf{100.00}_{\pm0.00}$ | $60.13_{\pm6.82}$ | $94.89_{\pm1.39}$ |
| **W+R** | $\mathbf{100.00}_{\pm0.00}$ | $\mathbf{100.00}_{\pm0.00}$ | $55.71_{\pm11.20}$ | $94.17_{\pm1.58}$ | $\mathbf{100.00}_{\pm0.00}$ | $\mathbf{100.00}_{\pm0.00}$ | $59.89_{\pm15.18}$ | $95.09_{\pm2.14}$ |
| **W+I** | $\mathbf{100.00}_{\pm0.00}$ | $\mathbf{100.00}_{\pm0.00}$ | $56.53_{\pm12.23}$ | $94.38_{\pm1.70}$ | $\mathbf{100.00}_{\pm0.00}$ | $\mathbf{100.00}_{\pm0.00}$ | $62.42_{\pm5.65}$ | $95.26_{\pm1.08}$ |
| **I+R** | $\mathbf{100.00}_{\pm0.00}$ | $99.61_{\pm0.67}$ | $60.10_{\pm11.97}$ | $94.54_{\pm1.45}$ | $\mathbf{100.00}_{\pm0.00}$ | $\mathbf{100.00}_{\pm0.00}$ | $61.10_{\pm8.46}$ | $94.80_{\pm1.28}$ |
| **W+R+I** | $\mathbf{100.00}_{\pm0.00}$ | $\mathbf{100.00}_{\pm0.00}$ | $\mathbf{60.62}_{\pm8.42}$ | $\mathbf{94.88}_{\pm1.75}$ | $\mathbf{100.00}_{\pm0.00}$ | $\mathbf{100.00}_{\pm0.00}$ | $\mathbf{66.16}_{\pm10.89}$ | $\mathbf{95.83}_{\pm1.50}$ |

Table 4: Action detection results with mmWave (**W**), RGB (**R**), IMUs (**I**), and their combinations. We report the mean and standard deviation of $m$AP averaged across 3 splits under our setting 2 (**S2**).

radar signals by 1.9% and 6.4%, respectively. Under **P2**, both IMUs data and RGB frames perform equally well with improved $m$AP (around 95%). The RGB frames achieve a major improvement when evaluated under **P2**. It is interesting to cross reference the results of HPE and action detection. While RGB frames have lower joint errors under **S2** and **P1**, they have slightly worse results on action detection. On the other hand, IMUs data perform consistently well on action detection in **P1** and **P2**.

Further combining the modalities results in a noticeable performance boost. It is probably not surprising that using all three modalities yields the best results, outperforming the best single modality by 1.4% (**P1**) and 0.8% (**P2**) in average mAP and with most gains in mAP under tIoU=0.95 (+7.1% for **P1** and +6.0% for **P2**). Fusing any of the two modalities leads to improved performance than the best of the constituting modality, except the combination of IMUs+RGB under **P2**. These results demonstrate a first step towards multi-modal learning with our dataset.

mmWave radar is less invasive than IMU sensors and offers better privacy than RGB cameras. While in its infancy for human sensing, this modality presents an emerging solution for home-based health monitoring. Part of our goal in this paper is to explore mmWave radar for human sensing by comparing its performance to other common modalities. The results indicate that mmWave radar leads to compelling performance for both human pose estimation and action localization. While its results are worse than those with RGB cameras or IMU sensors, mmWave radar might still be preferred for privacy-sensitive applications.

## 5   Ethics Statement

The human subject studies reported in this paper was reviewed and approved by the IRB committee at the University of Wisconsin-Madison. Each participant was informed about the research project and signed the consent form. The data has been de-identified with facial information blurred in all videos and participant ID anonymized, and made publicly available to facilitate future research.

To the best of the authors' knowledge, this work does not disadvantage any person directly. The authors do acknowledge that any pose estimation and activity recognition method can potentially be used with malicious intent, such as tracking user movements. If the human pose estimation/human activity understanding algorithms are directly used to make decisions for patients, potential failures in the classification would affect the users' quality of life. Therefore, the data and insights on patient activity must be verified by health professionals before making any decisions.

## 6   Conclusion and future work

In this paper, we proposed *mRI*— a multi-modal 3D human pose estimation dataset of rehabilitation exercises performed by 20 subjects, consisting of more than 160k synchronized frames. *mRI* combines mmWave, RGB-D, and IMUs as sensing modalities, and thus provides the most comprehensive benchmark to date for pose estimation and action detection. We described the creation of our dataset and demonstrated extensive benchmarks using our dataset. Our results help to understand the advantages of individual sensing modalities in the context of home-based health monitoring. We hope that *mRI* can catalyze the research including but not limited to pose estimation, multi-modal learning, and action understanding, thus facilitating critical applications in healthcare. We envision a variety of meaningful future work leveraging our dataset, drawing attention from communities including machine learning, computer vision, wearable computing, multi-modal sensing, and healthcare.

## Acknowledgments and Disclosure of Funding

This research was funded by NSF CAREER award CNS-2114499.

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
