# OpenReview forum: "mRI: Multi-modal 3D Human Pose Estimation Dataset using mmWave, RGB-D, and Inertial Sensors"
_NeurIPS.cc/2022/Track/Datasets_and_Benchmarks — NeurIPS 2022 Datasets and Benchmarks _

### Official Review · Reviewer_mXzT · 2022-07-04
**Rejection: Weak Experiments and Bad Visualization.**

**Rating:** 6
**Confidence:** 5
**Clarity:** Yes.

**Strengths:**

A new dataset for 3D human pose estimation on home-based health monitoring, contains the data from multiple sensors including mmWave, RGB-D, and 9 Inertial Sensors.

**Weaknesses:**

# Poor Evidence/Experiments:
The experiments are too weak so that it does not prove this multi-modal 3D dataset to be useful and needed.
* The dataset paper does not explore how to fuse data from different sensors to get better results, so that can prove multiple sensors are better than any single one.
* A very important experiment is to prove that this dataset contains all three devices (RGB-D, IMU, mmWave) are better than two devices (RGB-D and IMU); Or Considering privacy, the paper does not prove why this extra mmWave sensor or IMU sensor is useful compared with previous existing datasets (MoVi[11] or MARS[2]) without using RGB data.

# Not Clear Comparison Table 1
If the dataset is already released, it would be much better to count all required numbers for the whole table.
For example:
* MoVi[11] has released the dataset, it has more subjects, more sequences, and more action numbers, it potentially has more frames, it would be much better if the author can calculate the number instead of "not listing".

# Not Obvious Visualization
The dataset is for **"3D"** human pose estimation, it should not use only one 2D screenshot for visualization of 3D skeletons, which is quite ambiguous. It should visualize multiple views (bird-view, side-view) of the results or add extra projection on x-z and x-y planes.

The visualization for mmWave is also too bad, does not get how to generate that "random" points for that person from the mmWave radar.
And the visualization of this dataset are not obvious.

# What's the distance between the captured human and the kinect V2?

**Additional Feedback:**

Nope.

**Correctness:**

Yes. It is a dataset paper. And for those insufficient claims, see in Weaknesses.


**Documentation:**

Yes.

**Ethics:**

No.

**Relation To Prior Work:**

Not that clear. See in Weakness.

**Summary And Contributions:**

This paper collect a multi-modal 3D human pose estimation dataset with mmWave, RGB-D, and 9 Inertial Sensors to enable future research on multiple modalities on home-based health monitoring.

---

> ### Author Response · Authors · 2022-08-20
> **Thanks for the constructive feedback**
>
> We thank the reviewer for the constructive feedback. It helps us improve the paper tremendously.
>
> Comment 1.1: Poor Evidence/Experiments:
>
> Response:
>
> Our key contribution is a multi-modal dataset for human pose estimation and activity understanding in a home-based health monitoring context. Our experiments aim to evaluate individual modalities in terms of performance, privacy, and invasiveness. While we agree that multi-modal fusion is an interesting direction, it is not the focus of this paper.
>
> Per the reviewer’s request, we conducted additional experiments on action localization using multiple modalities. Specifically, we concatenate the 3D joint data from individual modalities at every time step, and send the resulting sequence into the action localization model. The results of combining different modalities are shown in the table below.
>
> |             |             |             |             |            | mAP↑ |             |             |             |            |   |
> |-------------|:-----------:|:-----------:|:-----------:|:----------:|:----:|:-----------:|:-----------:|:-----------:|------------|---|
> |             |             |             |  Protocol 1 |            |      |             |             |  Protocol 2 |            |   |
> | Modality    | tIoU=0.50   | tIoU=0.75   | tIoU=0.95   | average    |      | tIoU=0.50   | tIoU=0.75   | tIoU=0.95   | average    |   |
> | mmWave (W)  | 98.22±3.08  | 97.59±4.17  | 29.02±6.31  | 87.04±4.89 |      | 99.00±1.73  | 97.21±2.41  | 31.11±15.34 | 87.55±3.61 |   |
> | RGB (R)     | 100.00±0.00 | 99.14±0.75  | 44.80±10.55 | 91.56±2.08 |      | 100.00±0.00 | 100.00±0.00 | 59.87±8.12  | 95.07±1.46 |   |
> | IMUs (I)    | 100.00±0.00 | 100.00±0.00 | 53.55±12.39 | 93.46±2.30 |      | 100.00±0.00 | 100.00±0.00 | 60.13±6.82  | 94.89±1.39 |   |
> | W+R         | 100.00±0.00 | 100.00±0.00 | 55.71±11.20 | 94.17±1.58 |      | 100.00±0.00 | 100.00±0.00 | 59.89±15.18 | 95.09±2.14 |   |
> | W+I         | 100.00±0.00 | 100.00±0.00 | 56.53±12.23 | 94.38±1.70 |      | 100.00±0.00 | 100.00±0.00 | 62.42±5.65  | 95.26±1.08 |   |
> | I+R         | 100.00±0.00 | 99.61±0.67  | 60.10±11.97 | 94.54±1.45 |      | 100.00±0.00 | 100.00±0.00 | 61.10±8.46  | 94.80±1.28 |   |
> | W+R+I       | 100.00±0.00 | 100.00±0.00 | 60.62±8.42  | 94.88±1.75 |      | 100.00±0.00 | 100.00±0.00 | 66.16±10.89 | 95.83±1.50 |   |
>
> Further combining multiple modalities results in a noticeable performance boost. Fusing any of the two modalities leads to better performance than the best of the constituting modality, except the combination of IMUs+RGB under protocol 2. Using all three modalities indeed yields the best results, outperforming the best single modality by 1.4% (in protocol 1) and 0.8% (in protocol 2) in average mean average precision (mAP) and with most gains in mAP under tight temporal intersection over union (tIoU) threshold of 0.95 (+7.1% for protocol 1 and +6.0% for protocol 2).
>
> These results demonstrate a first step towards multi-modal learning with our dataset. We have updated the paper with these new results. See Table 4 and Sec 4.2 of the revised paper.
>
> Comment 1.2: Not Clear Comparison Table 1
>
> Response:
>
> We calculated the frames for MoVi and filled that in the table. In addition, we’ve updated the table and filled as many metrics as possible for other datasets.
>
> Comment 1.3: Not Obvious Visualization
>
> Response:
>
> To better demonstrate 3D poses, we have added new visualization of multiple views to Sec 4.1 and supplement A.5 of the revised manuscript. The full 3D visualization is also updated on the project website (https://sizhean.github.io/mri). The tools for generating the plots will also be shared.
> We also realized that more background information about mmWave imaging would be helpful. Therefore, the revised manuscript describes the generation of point clouds in Supplementary A.4 mmWave Imaging. For more details, please refer to [1,2,3].
>
> [1] S. Rao. Introduction to mmwave sensing: Fmcw radars. Texas Instruments (TI) mmWave Training Series, 2017.
>
> [2] Texas Instruments. mmWavetutorial. https://www.ti.com/lit/pdf/swra553 accessed 29 Sep. 2020.
>
> [3] Texas Instruments. mmWavefundamentals. https://www.ti.com/lit/spyy005 accessed 8 Apr. 2021.
>
> Comment 1.4: What's the distance between the captured human and the kinect V2?
>
> Response:
>
> The subject is standing in the position 2.4m away from the camera. This was mentioned in the original paper- Section 3.1 and figure 2(f).
>
> A small correction: In <summary and contribution part> of your comments '9 inertial sensors' are mentioned. However, we only used six IMUs and claimed so.
>
> We hope our responses adequately address all comments and welcome any further questions during the discussion.

---

> ### Author Response · Authors · 2022-08-28
> **Have we addressed your comments?**
>
> Hi Reviewer mXzT,
>
> Thank you again for your insightful feedback. Have we adequately addressed your questions and concerns? Or is there anything else we can answer for you? As the discussion is closing soon, we would like to take the last chance to answer any further concerns you have.

---

### Official Review · Reviewer_CPkq · 2022-07-27
**A rich multi-modal dataset for pose estimation and action detection**

**Rating:** 6
**Confidence:** 4
**Correctness:** Yes
**Clarity:** Yes, the paper is well written.

**Strengths:**

1. It is the first dataset to have all three modalities (IMU, RGB-D and mm Wave) that are synchronized. This is an important contribution that could be useful for training models with multi-modalities for pose estimation and action detection. It can also be useful for pretraining action detection/pose estimation models using multi-modal self-supervised learning.

2. The dataset also annotations for action labels, 2D key points and 3D key points. These annotations can be useful for training supervised models. These models can be useful starting points which can be further finetuned to perform action recognition or pose estimation tasks.

3. In comparison to other pose estimation datasets, this dataset has the most number of synchronize frames. This make the dataset suitable for training models for human action understanding.

4. The dataset is relatively well documented.


**Weaknesses:**

There are few concerns. If authors can resolve them, it would increase the quality of submission.
1. Quality of the labels: How many annotators annotated the dataset (action labels)? If there were multiple annotators, what was their inter-rater reliability coefficient?

For the key points annotations, how were the model generated key points validated? Were they validated in ad-hoc manner or some proper statistical sampling performed to ensure robust validation?

 What are chances that key point annotations generated by the model are noisy? Is there a way to quantify it (using some kind of sampling and manual inspection)?

2. To ensure reproducibility of the results and benchmarking, it would be advisable to share the train and test splits for each protocol (S1 and S2).

3. How many trials were performed by each subject? Were  subjects free to do any of the 12 movements at any time or was there an experimental protocol followed that instructed the subjects what to do?

4. Currently, the dataset has only young and healthy adults. It would be ideal if the diversity of the data is increased by including olde/elder subjects and possibly subjects with some kind of motor impairment.

5.  To ensure reproducibility of the results, it would be advisable to release the code for training the benchmark models.

6. Since this the first dataset which synchronously captures three modalities, it would be interesting to see how the performance for pose estimation and action detection improves if the model is trained on multiple modalities.

**Additional Feedback:**

None

**Documentation:**

Dataset is well documented. Video data (RGB-D) is yet to be processed. As per the authors, it would be available by the conference date.The code for reproducing benchmarks is missing. It would be nice to share it for reproducibility purposes.


**Ethics:**

No.

**Relation To Prior Work:**

Yes

**Summary And Contributions:**

The paper introduces a new multi-modal dataset which can be used for training models to estimate 3D human pose. The dataset contains three modalities namely: IMU sensors, RGB-D, and mm Wave. The captured data are annotated with 2D key points, 3D key points/skeleton and actions (start and end timestamp) performed by the subject. The paper also demonstrates how different modalities perform the tasks of human pose estimation and action detection.

---

> ### Author Response · Authors · 2022-08-20
> **Thanks for the constructive feedback**
>
> We thank the reviewer for the constructive feedback. It helps us improve the paper tremendously.
>
> Response to point 1:
>
> One annotator annotated all videos so inter-rater reliability is not applicable here.  We are working on having another annotator annotate the subset of the videos. The result will be updated in this comment before the deadline.
>
> To validate the reliability of the obtained 3D joints, we conduct additional experiments and report the reprojection error of the derived 3D joints. This error is defined as the distance between the 2D projections of the derived 3D joints and the human-annotated 2D keypoints. Lower reprojection errors thus indicate more accurate 3D poses. To this end, we randomly sample 50 video frames from our dataset, manually annotate the 2D keypoints for each frame, and calculate reprojection error, following [1, 2]. The mean absolute percentage error (MAPE) is 1.5%, while the percentage of correct keypoints thresholded at 50% of the head segment length (PCKh) is 98.92. We have included the results in Sec 3.2 of the revised paper. We provided more details and visualization in the Supplement A.2.
>
> [1] Kanazawa, Angjoo, et al. "End-to-end recovery of human shape and pose." Proceedings of the IEEE conference on computer vision and pattern recognition. 2018.
>
> [2] Andriluka, Mykhaylo, et al. "2D human pose estimation: New benchmark and state of the art analysis." Proceedings of the IEEE Conference on Computer Vision and Pattern Recognition. 2014.
>
>
> Response to point 2:
>
> S2 is shared in supplementary A.3. Random seeds for S1 will be shared in the code.
>
> Response to point 3:
>
> Each subject repeats every type of movement for about one minute. The order of the movements are fixed. This description is now added to Section 3.2 of the revised paper.
>
> Response to point 4:
>
> We thank the reviewer for this suggestion. We believe that to understand the movement of patients, it is an important first step to understand the movement of healthy people, which is the main focus of this paper. We plan to consider subjects with motor impairments in future work in collaboration with medical experts.
>
> Response to point 5:
>
> All code will be made publicly available.
>
> Response to point 6:
>
> This is also mentioned by another reviewer. Our key contribution is a multi-modal dataset for human pose estimation and activity understanding in a home-based health monitoring context. Our experiments aim to evaluate individual modalities in terms of performance, privacy, and invasiveness. While we agree that multi-modal fusion is an interesting direction, it is not the focus of this paper.
> Per the reviewer’s request, we conducted additional experiments on action localization using multiple modalities. Specifically, we concatenate the 3D joint data from individual modalities at every time step and send the resulting sequence into the action localization model. The results of combining different modalities are shown in the table (Please refer to reponses to Reviewer mXzT or Section 4.2 in paper).
>
> Further combining multiple modalities results in a noticeable performance boost. Fusing any of the two modalities leads to better performance than the best of the constituting modality, except the combination of IMUs+RGB under protocol 2. Using all three modalities indeed yields the best results, outperforming the best single modality by 1.4% (in protocol 1) and 0.8% (in protocol 2) in average mean average precision (mAP) and with most gains in mAP under tight temporal intersection over union (tIoU) threshold of 0.95 (+7.1% for protocol 1 and +6.0% for protocol 2).
>
> These results demonstrate the first step towards multi-modal learning with our dataset. We have updated the paper with these new results. See Table 4 and Sec 4.2 of the revised paper.
>
> We hope our responses adequately address all comments and welcome any further questions during the discussion.

---

> > ### Author Response · Authors · 2022-08-25
> > **Updated results on validating the inter-rater reliability (point1) and code availability (point5)**
> >
> > To validate the inter-rater reliability, we had another person annotate five videos and calculate the linear Cohen's kappa coefficient [1] between two annotators' results. The coefficient is 0.98, which indicates that two annotations are highly aligned.
> >
> > We shared the code of action localization. Code of pose estimation will be shared with the RGB data later.
> >
> > We thank the reviewer for the insightful feedback.
> >
> > [1] McHugh ML. Interrater reliability: the kappa statistic. Biochem Med (Zagreb). 2012;22(3):276-82. PMID: 23092060; PMCID: PMC3900052.

---

> ### Author Response · Authors · 2022-08-28
> **Have we addressed your comments?**
>
> Hi Reviewer CPkq,
>
> Thank you again for your insightful feedback. Have we adequately addressed your questions and concerns? Or is there anything else we can answer for you? As the discussion is closing soon, we would like to take the last chance to answer any further concerns you have.

---

### Official Review · Reviewer_C3ax · 2022-07-28
**mRI: Multi-modal 3D Human Pose Estimation Dataset using mmWave, RGB-D, and Inertial Sensors**

**Rating:** 7
**Confidence:** 3

**Strengths:**

It is valuable to have multi-modal datasets, and facilitating at-home rehabilitation with personalized feedback is laudable.  The dataset contains a good mix of sensors that may facilitate cross-modal analysis.  The benchmarks address the common tasks of joint reconstruction and activity recognition.

**Weaknesses:**

A few questions/comments that came to mind are summarized below:

- The paper described the dataset as comprising over 5 million frames, but this does not seem like a particularly useful size metric; I think duration would be more meaningful.  I'm not actually sure if the paper mentioned metrics such as duration (per subject, per activity, etc.), repetitions of activities, etc.  These should be added to better describe the dataset.  The number of synced frames was reported, which comes closer to measuring duration, but it is still unclear.

- Table 1 says that some papers report synchronized frames and others total frames, but it's not clear which is which in the table (so that column may be an unfair comparison).

- It seems like the optimization method is used to estimate body poses from the data, and then those are used as ground truth for evaluating the other pipelines that also aim to estimate joint positions?  The underlying assumption thus seems to be that the optimization method is much more accurate than the learning pipelines, even though its accuracy cannot be thoroughly evaluated?  Some discussion about the possible implications of this may be good to add.  (Sorry if I'm missing that there was another source of ground truth for the benchmarks.)

- A bit more discussion about why the two cameras are next to each other may be helpful.  I had originally assumed that they would provide different vantage points to avoid occlusions, but now I suspect they are just to provide stereo data?  This may be redundant with the depth data though, unless the aim is to explore stereo vision on its own (perhaps using depth data as ground truth)?

- It was mentioned that the six IMUs capture richer pose representations than the mmWave radar; some more elaboration on that may be useful, since it's not immediately clear given that the radar is providing a point cloud encompassing the entire person if I am understanding it correctly.

- Some more information about the activity detection approach may be helpful, such as the duration of the segments being classified and whether it can operate in a real-time streaming fashion or assumes segmented examples.  Thoughts on why certain modalities are better than others for certain activities may be useful as well.  A confusion matrix may also be interesting.

- The ethics section should probably be in the main paper portion before the conclusion?  Also, the last sentence saying that the approach "does not leverage any bias in the datasets used for validation" was a bit unclear.  There is undoubtedly some bias in the dataset, such as from the recruitment pool yielding similar subject backgrounds or abilities, so mentioning some possible sources (or at least words of caution) may be helpful.

- Section 3.1 mentions that the slowest sensor is used as the base timestamp, and then the closest sample from each sensor is found.  Does this imply that all sensors are being resampled to the lowest sampling rate?  I hope all data is provided at their original sampling rates in the dataset as well, rather than that data being discarded?  Downsampling may be useful for some applications, but can be a significant drawback for others.  Update: I see that the GitHub includes raw data, but it would be helpful to mention this in the paper as well (along with more information about the dataset structure and formats).

- Left and right may be swapped in section 3.2 when describing Figure 3?

**Additional Feedback:**

Congrats on your experiments!

**Clarity:**

The paper is generally well written, although there are typos and grammatical adjustments to address.

**Correctness:**

The evaluation methods seem reasonable, especially the inclusion of leave-one-subject-out validation.  It is a bit hard to evaluate the dataset construction as it is not yet available and little information about its format is provided.  The data collection procedure sounds reasonable though.

**Documentation:**

More information about the dataset, file formats, usage instructions, hosting plans, access, licenses, etc. would be beneficial.  There is some more information on the GitHub at the moment that addresses these topics, but including them in the paper and expanding upon them would make it more comprehensive.

**Ethics:**

The ethics section raises most of the major concerns.  See above comment about potentially expanding on the point about bias. Adding something about how future users should not actively try to identify subjects may also be good to include (it's great to obscure the videos, but it may still be possible to deduce information about the subjects if someone tried, so I've seen other datasets include language to that effect).

**Relation To Prior Work:**

The related work section nicely discusses relevant areas including human pose detection, related datasets, and rehabilitation.  The table comparing datasets seems comprehensive and helpful.

**Summary And Contributions:**

The paper presents a multi-modal dataset of mmWave, RGB-D, and IMU readings during rehabilitation-focused movements in healthy subjects.  It spans 20 subjects and includes 12 labels (10 motions/exercises and 2 free-form periods).  Joint angles are determined via an optimization procedure, and activity labels are provided.  The paper also presents results about extracting joint angles from each sensor subset, and about classifying activities based on joint angles.

---

> ### Author Response · Authors · 2022-08-20
> **Thanks for the constructive feedback**
>
> We thank the reviewer for the constructive feedback. It helps us improve the paper tremendously.
>
> Response to point1:
>
> Each subject repeats every type of movement for about one minute. This description is now added to Section 3.2 of the revised paper.
>
> Response to point2:
>
> We’ve updated the table and filled as many metrics as possible. In addition, we discard the ‘total frames’ metrics per request, and report ‘synchronized frames’ for all datasets.
>
> Response to point3:
>
> Yes, the reviewer’s understanding is correct. Without using a MoCap system, we design an optimization scheme to derive 3D poses for model training and evaluation. The optimization enforces equal bone length constraints and promotes temporal smoothness in the video. The resulting poses are considered as “pseudo” ground-truth. This paradigm of obtaining 3D poses from multiple cameras and using them for evaluation is also considered in other works, such as:
>
> [1] Karashchuk, Pierre, et al. "Anipose: a toolkit for robust markerless 3D pose estimation." Cell reports 36.13 (2021): 109730.
>
> [2] Luo, Yiyue, et al. "Intelligent carpet: Inferring 3d human pose from tactile signals." Proceedings of the IEEE/CVF Conference on Computer Vision and Pattern Recognition. 2021.
>
> [3] Cai, Zhongang, et al. "HuMMan: Multi-Modal 4D Human Dataset for Versatile Sensing and Modeling." European Conference on Computer Vision. 2022.
>
> To validate the reliability of the obtained 3D joints, we conduct additional experiments and report the reprojection error of the derived 3D joints. This error is defined as the distance between the 2D projections of the derived 3D joints and the human-annotated 2D keypoints. Lower reprojection errors thus indicate more accurate 3D poses. To this end, we randomly sample 50 video frames from our dataset, manually annotate the 2D keypoints for each frame, and calculate reprojection error, following [4, 5]. The mean absolute percentage error (MAPE) is 1.5%, and the percentage of correct keypoints thresholded at 50% of the head segment length (PCKh) is 98.92. We have included the results in Sec 3.2 of the revised paper. More details and visualization can be found in the Supplement A.2.
>
> [4] Kanazawa, Angjoo, et al. "End-to-end recovery of human shape and pose." Proceedings of the IEEE conference on computer vision and pattern recognition. 2018.
>
> [5] Andriluka, Mykhaylo, et al. "2D human pose estimation: New benchmark and state of the art analysis." Proceedings of the IEEE Conference on Computer Vision and Pattern Recognition. 2014.
>
> Response to point4:
>
> Yes, the two cameras are conceptually similar to a stereo pair. They are used to derive the “pseudo” ground truth of 3D joints for model training and evaluation. We recognize that this is indeed redundant with the depth maps. Our current work has not thoroughly explored the depth maps due to issues in synchronization, calibration and image alignment. As a future work, we plan to investigate the depth modality for human pose estimation and action localization tasks.
>
> Response to point5:
>
> We realized that more background information about mmWave imaging would be helpful. Therefore, the revised manuscript describes the generation of point clouds in Supplementary A.4 mmWave Imaging. For more details, please refer to [1,2,3].
>
> [1] S. Rao. Introduction to mmwave sensing: Fmcw radars. Texas Instruments (TI) mmWave Training Series, 2017.
>
> [2] https://www.ti.com/lit/pdf/swra553
>
> [3] https://www.ti.com/lit/spyy005
>
> mmWave radar measures rough body shape and motion based on echos, and thus the reconstruction of body pose can be challenging. In contrast, body-worn IMU sensors directly measure the angular rate and acceleration of body parts, from which the body pose can be readily derived.
>
> Response to point6:
>
> Action detection is a localization problem with untrimmed videos (structure prediction) rather than the classification of segmented videos. Specifically, given an input untrimmed video, temporal action localization seeks to predict a set of action instances of varying size. Each instance is defined by its onset, offset, and action labels. Therefore, no confusion matrix can be applied here. We have added a more detailed description about action localization in Sec 4.2.
>
> Response to point7:
>
> We have moved the ethics section to the main paper and revised the sentence. Please refer to Section 5 of the revised paper.
>
> Response to point8:
>
> Our synchronization resamples all data to the lowest sampling rate across the modalities. Both the raw data and the synchronized version are released as part of our dataset. This is now clarified in the supplement A.3. We will keep improving the documentation in the github page.
>
> Response to point9:
>
> We mirrored the figures to avoid the confusion but it looks like it created more confusion. Now we update the figure to match the texts.
>
>
> We hope our responses adequately address all comments and welcome any further questions during the discussion.

---

> > ### Author Response · Authors · 2022-08-25
> > **Updated results on validating the learning pipeline VS optimization-based joints (point3)**
> >
> > To further compare the 3D joints quality from the learning pipeline and optimization-based joints, we reproject the 2D keypoints using 3D joints from the RGB model's inference and optimization-based method. Then we calculate the error between those 2D keypoints and human-annotated keypoints (as mentioned in the previous response). The MAPE between human annotation and learning pipeline is 2.8%, while the optimization-based method is 1.5%. We can observe that joints from the optimization-based method are more reliable than the learning pipeline.
> >
> > We thank the reviewer for the insightful feedback.

---

> > ### Comment · Reviewer_C3ax · 2022-09-02
> > **Response to responses**
> >
> > Thank you for the thoughtful responses and revision!

---

> ### Author Response · Authors · 2022-08-28
> **Have we addressed your comments?**
>
> Hi Reviewer C3ax,
>
> Thank you again for your insightful feedback. Have we adequately addressed your questions and concerns? Or is there anything else we can answer for you? As the discussion is closing soon, we would like to take the last chance to answer any further concerns you have.

---

### Official Review · Reviewer_S6jw · 2022-07-28

**Rating:** 6
**Confidence:** 4
**Correctness:** Yes
**Clarity:** Yes

**Strengths:**

+ Paper is well written. The dataset statistics and data collection details are clearly mentioned.
+ It is the first dataset with the most comprehensive set of sensing modalities, including RGB, depth, IMU, and mmWave.xnww

**Weaknesses:**

- This work proposes a multi-modal dataset but only evaluates the standard methods for each modality separately. As shown in Table 3, while using only the IMU obtains accurate results (40.9 MPJPE and 28.4 PA-MPJPE), it is not clear whether using multiple modalities is beneficial and necessary.

- The 3D keypoints are labeled with the RGB-based 2D pose estimation method with triangulation, which is not robust and vulnerable to occlusion. Therefore, the ground-truth 3D poses are not accurate. To ensure accurate 2D pose estimation, the RGB image should be captured in a controlled environment, while there are already lots of indoor datasets that provide 3D pose annotations.

- In table 3, the methods using mmWave, RGB, and IMUs are trained in different settings. Thus the direct comparison is not fair.



**Additional Feedback:**

I suggest the authors report the results of the RGB-based 3D pose estimator that is fine-tuned in mRI.

**Documentation:**

Yes

**Ethics:**

Yes

**Relation To Prior Work:**

Yes

**Summary And Contributions:**

This paper describes the mRI dataset, a multi-modal 3D human pose estimation dataset with mmWave, RGB-D, and Inertial Sensors. mRI consists of over 5 million frames from 20 subjects performing rehabilitation exercises. The authors give benchmarks for the dataset in two categories - 3D human pose estimation and action detection. 3D Human Pose Estimation has two evaluation protocols - direct 3D pose evaluation and cross-subject generalization.

The main contribution of this dataset is that it supports multiple sensing modalities. Besides, mRI focuses on healthcare and only requires low-power and low-cost devices.

---

> ### Author Response · Authors · 2022-08-20
> **Thanks for the constructive feedback**
>
> We thank the reviewer for the constructive feedback. It helps us improve the paper tremendously.
>
> Response to point1:
>
> This is also mentioned by another reviewer. Our key contribution is a multi-modal dataset for human pose estimation and activity understanding in a home-based health monitoring context. Our experiments aim to evaluate individual modalities in terms of performance, privacy, and invasiveness. While we agree that multi-modal fusion is an interesting direction, it is not the focus of this paper. Per the reviewer’s request, we conducted additional experiments on action localization using multiple modalities. Specifically, we concatenate the 3D joint data from individual modalities at every time step and send the resulting sequence into the action localization model. The results of combining different modalities are shown in the table (Please refer to reponses to Reviewer mXzT or Section 4.2 in paper).
>
> Further combining multiple modalities results in a noticeable performance boost. Fusing any of the two modalities leads to better performance than the best of the constituting modality, except the combination of IMUs+RGB under protocol 2. Using all three modalities indeed yields the best results, outperforming the best single modality by 1.4% (in protocol 1) and 0.8% (in protocol 2) in average mean average precision (mAP) and with most gains in mAP under tight temporal intersection over union (tIoU) threshold of 0.95 (+7.1% for protocol 1 and +6.0% for protocol 2).
>
> These results demonstrate the first step towards multi-modal learning with our dataset. We have updated the paper with these new results. See Table 4 and Sec 4.2 of the revised paper.
>
> Response to point2:
>
> We agree that the “ground truth” 3D poses are not necessarily perfect.
>
> To validate the reliability of the obtained 3D joints, we report the reprojection error of the derived 3D joints by comparing their 2D projections to human annotated 2D keypoints. Specifically, we randomly sample 50 video frames from our dataset, manually annotate the 2D keypoints for each frame, and calculate the error between the projected 3D joints and the annotated 2D keypoints, following [1,2]. The mean absolute error (MAE) is 9.7pixels, mean absolute percentage error (MAPE) is 1.5%, while the percentage of correct keypoints thresholded at 50% of the head segment length (PCKh) is 98.92. More details and visualization can be found in the Supplement A.2.
>
> [1] Kanazawa, Angjoo, et al. "End-to-end recovery of human shape and pose." Proceedings of the IEEE conference on computer vision and pattern recognition. 2018.
>
> [2] Andriluka, Mykhaylo, et al. "2d human pose estimation: New benchmark and state of the art analysis." Proceedings of the IEEE Conference on computer Vision and Pattern Recognition. 2014.
>
> Since the dataset targets home-based rehabilitation, we have a controlled indoor environment. This means we do not consider occlusion from objects/other subjects. We agree that there are already lots of indoor datasets that provide 3D pose annotations. However, our dataset is unique in two aspects:
>
> 1) We collect synchronized data from emerging sensing modalities including IMUs and mmWave point cloud. For the first time, standardized mmWave-based human pose estimation with proper video references and keypoints annotations are proposed.
>
> 2) The dataset is health-focused and we select ten rehabilitation movements suggested by the clinical expert. It  helps to understand the advantages of individual sensing modalities in the context of home-based health monitoring. We hope that mRI can catalyze the research including but not limited to pose estimation, multi-modal learning, and action understanding, thus facilitating critical applications in healthcare.
>
> Response to point3:
>
> We are working on finetuning the RGB-based 3D pose estimator on mRI such that they are under the same settings (In our understanding, settings here mean mmWave/IMU are trained from scratch while RGB are not finetuned/trained with. Please correct us if not). Will update the results in the response.
>
> Response to additional feedback:
>
> Please see response to point3.
>
> We hope our responses adequately address the rest of the comments and welcome any further questions during the discussion.

---

> > ### Author Response · Authors · 2022-08-25
> > **Updated results on RGB-based 3D pose estimator (point3)**
> >
> > We have conducted new experiments regarding re-training the RGB-based 3D pose estimator per reviewer's request. We performed re-training instead of fine-tuning as the reviewer suggested since the 3D keypoints format used in VideoPose3D (Human3.6M) is different from our triangulated 3D joints (COCO). We follow the same settings with other modalities (Split 1/2 and Protocol 1/2), and the results are shown below:
> >
> > |              | Protocol 1 |               | Protocol 2 |               |
> > |--------------|:----------:|:-------------:|:----------:|:-------------:|
> > | Setting      | MPJPE(mm)↓ | PA-MPJPE(mm)↓ | MPJPE(mm)↓ | PA-MPJPE(mm)↓ |
> > | Random (S1)  | 92.8±11.9  | 54.1±7.1      | 49.5±1.0   | 32.4±0.5      |
> > | Subject (S2) | 116.1±19.7 | 60.9±4.5      | 62.0±5.7   | 39.8±3.4      |
> >
> > We observe a significant improvement compared with only testing with the pre-trained model. RGB-trained 3D pose estimator is comparable to IMU-trained model in terms of the performance, in some cases (S2P2) even better. We really appreciate the constructive and insightful feedback.

---

> > ### Comment · Reviewer_S6jw · 2022-08-25
> > **Reply to the authors**
> >
> > Thanks for the feedback from the authors. The feedback resolved my concerns and I would like to raise my rating.

---

> > > ### Author Response · Authors · 2022-08-28
> > > **Reply to the reviewer**
> > >
> > > Hi Reviewer S6jw,
> > >
> > > We greatly enjoy the discussion with you and appreciate the rating raise. Thank you very much for the supportive feedback.

---

### Official Review · Reviewer_7Tmu · 2022-07-30
**A Multi-modal 3D Human Pose Estimation Dataset with Detailed Documentation and Extensive Benchmarking**

**Rating:** 7
**Confidence:** 5

**Strengths:**

The authors managed to identify the gap from prior works, and thus proposed their dataset for 3D human pose estimation. This dataset offers some beneficial properties i.e. low cost, privacy, and low computational power.

Their multi-modality also allows for more creative adaptation of application using this dataset. The dataset focuses on healthcare rehabilitation, making it having high relevance to such usage versus some other generic datasets.

Apart from comparing their dataset against several closely related prior works, the authors also conducted some extensive benchmark using their dataset.

**Weaknesses:**

In Line 194 - 196, the authors mentioned about the detection of 2D keypoints using HRNet, and they confirmed that these detected 2D keypoints are reasonably accurate. However, there is no details about their approach of confirming the accuracy of these 2D keypoints. These details could be important as they are directly affecting the generated dataset.

**Additional Feedback:**

Since the authors stated their benchmark results and some claims on different settings / modality that showed different performances, it will be important that the code implementation / trained models / configurations are given in the landing page / Github site of the dataset for reproduction (for the purpose of evaluation by the reviewers or other researchers).

**Clarity:**

The sections are well-organized, and easy for readership. Each point is clearly written and ideas were presented with clarity. Arguments were properly elaborated, with supportive descriptions.

**Correctness:**

The dataset is documented and organized in a standard manner.

The benchmarks were evaluated using standard metric.

**Documentation:**

The authors gave sufficient details on data collection and how they were organized. Some descriptions about the dataset accessibility / availability were documented. They mentioned about extra work on the dataset(due to privacy-preserving protocol), and they will only be publishing the dataset publicly once this is solved.

**Ethics:**

The authors have stated the ethical concerns clearly, and it seems that their work is not causing any potential negative impacts to the society, or the involved subjects.

**Relation To Prior Work:**

Discussion of prior works was given, and the authors listed some major relevant datasets for comparison. The authors managed to point out their work differences as compared to the prior works.

**Summary And Contributions:**

The authors proposed a multi-modal 3D human pose estimation dataset, targeting rehabilitation exercises. The collected the dataset using 20 subjects. They combined mmWave, RGB-D, and IMUs as sensing modalities, which is different from most of the prior works that are largely focusing on RGB, Lidar and the likes.

They documented the dataset generation details, including methods of data collection and annotation. The authors provided benchmark of several models using their dataset for all of their modalities. They ended the paper with discussion on ethical aspects, conclusion and future work.

---

> ### Author Response · Authors · 2022-08-20
> **Thanks for the constructive feedback**
>
> We thank the reviewer for the constructive feedback. It helps us improve the paper tremendously.
>
> Response to point1:
>
> To validate the reliability of the obtained 3D joints, we conduct additional experiments and report the reprojection error of the derived 3D joints. This error is defined as the distance between the 2D projections of the derived 3D joints and the human-annotated 2D keypoints. Lower reprojection errors thus indicate more accurate 3D poses. To this end, we randomly sample 50 video frames from our dataset, manually annotate the 2D keypoints for each frame, and calculate reprojection error, following [1, 2]. The mean absolute percentage error (MAPE) is 1.5%, and the percentage of correct keypoints thresholded at 50% of the head segment length (PCKh) is 98.92. We have included the results in Sec 3.2 of the revised paper. More details and visualization can be found in the Supplement A.2.
>
> [1] Kanazawa, Angjoo, et al. "End-to-end recovery of human shape and pose." Proceedings of the IEEE conference on computer vision and pattern recognition. 2018.
>
> [2] Andriluka, Mykhaylo, et al. "2D human pose estimation: New benchmark and state of the art analysis." Proceedings of the IEEE Conference on Computer Vision and Pattern Recognition. 2014.
>
> Response to Additional Feedback:
>
> S2 is shared in supplementary A.3. Random seeds for S1 will be shared in the code. There are some additional work needed before we can share the data and code such as subject de-identification. We promise that all code and will be made publicly available by the conference date.
>
> We hope our responses adequately address all comments and welcome any further questions during the discussion.

---

> ### Author Response · Authors · 2022-08-28
> **Have we addressed your comments?**
>
> Hi Reviewer 7Tmu,
>
> Thank you again for your insightful feedback. Have we adequately addressed your questions and concerns? Or is there anything else we can answer for you? As the discussion is closing soon, we would like to take the last chance to answer any further concerns you have.

---

### Author Response · Authors · 2022-08-20
**Summary response to reviewers**

We sincerely thank all the reviewers for the constructive feedback. We are encouraged to learn that the reviewers found our dataset as “the first dataset with the most comprehensive set of sensing modalities,” (Reviewer CPkq, S6jw, and 7Tmu) offering “some beneficial properties such as low cost, privacy and low computational power” (Reviewer 7Tmu) and enabling “future research on multiple modalities on home-based health monitoring.” (Reviewer CPkq, C3ax, and 7Tmu)
To respond to the suggestions, we carefully went through all review comments, provided detailed responses to each reviewer, and revised our manuscript. Our revision of the manuscript is summarized as follows:
- Conducted additional experiments to quantify the 3D joint quality. This is done by annotating the 2D keypoints manually and calculating the reprojection error. Results and texts are updated in Section 3.2 and Supplement A.2.
- Conducted new experiments on action localization using a combination of multiple modalities. Texts and results are updated in Section 4.2.
- Updated figures of new 3D visualization in Section 4.1, Supplement A.5, and gif on the project page.
- Added a new section in Supplementary A.4. to describe the mmWave imaging.
- Updated Table 1 in the related work section and filled as many metrics as possible for other datasets.
- Shared data split in Supplementary A.3.
- Moved the ethics section to the main paper.
- Other minor text revisions suggested by reviewers.

In addition to the individual responses, we briefly describe the suggestions (S) shared among reviewers and our responses (R).

S: Evaluation of the quality of the 3D joints used for model training and evaluation.

R: Without using a MoCap system, the 3D joints used for training and evaluating our model are derived by (a) triangulation from two cameras and (b) an optimization that enforces the equal bone length constraint and temporal smoothness in the video. We have conducted additional experiments to measure the quality of the 3D joints. The 2D projections of the derived 3D joints are compared against human annotated 2D keypoints in video frames, and the reprojection error is reported. Our results show compelling quality of the 3D joints. The new experiments and results are now described in Sec 3.2 of the paper.

S: Demonstration of multi-modal fusion for human sensing.

R: Our key contribution is a multi-modal dataset for human pose estimation and activity understanding in a home-based health monitoring context. Our current experiments focus on evaluating individual modalities in terms of performance, privacy, and invasiveness. To further demonstrate multi-modal fusion, we conducted new experiments on action localization using a combination of multiple modalities. The results, as described in Sec 4.2 of the paper, suggest that multi-modal fusion indeed improves the performance of localizing and recognizing actions.

We would like to thank all reviewers again for their effort. The updated paper (revised texts are highlighted in blue typeface) is updated in the openreview portal and we will keep updating the paper and responses by the deadline. We hope our responses adequately address the shared concerns and welcome any further questions during the discussion.

---

> ### Author Response · Authors · 2022-08-25
> **Updated skeleton-based action localization code**
>
> We just update a sample code of our skeleton-based action localization in the github page. Pose estimation related code and RGB data will be released later. Stay tuned!

---

### Meta-Review · Area_Chair_D6wk · 2022-09-11

**Recommendation:** Accept
**Confidence:** 4

**Metareview:**

This paper presents a dataset of synchronized radar, intertial sensors and RGB-D images of healthy young people conducting rehabilitation exercises. The paper also demonstrates how this dataset can be used as a benchmark for human pose estimation and as a catalyst for multi-modal learning. Reviewers appreciated these aspects, while pointing out issues regarding presentation, bias and claims in the analysis. These issues appear well addressed in the revisions. I recommend accepting this paper to the NeurIPS 2022 Dataset and Benchmarks program.

---

### Decision · Program_Chairs · 2022-09-16

Accept